# The Multi-Level Influencing Factors of Internet Use Among the Elderly Population and Its Association with Mental Health Promotion: Empirical Research Based on Mixed Cross-Sectional Data

**DOI:** 10.3390/healthcare13151931

**Published:** 2025-08-07

**Authors:** Yifan Yang, Xinying He

**Affiliations:** School of Public Administration, Southwest Jiaotong University, No. 111, North Section 1, Second Ring Road, Chengdu 610031, China; yangswjtu@126.com

**Keywords:** internet use, mental health, elderly group, digital divide

## Abstract

**Background**: China is confronted with the dual challenges of deeply interwoven population aging and the digitalization process. The digital integration and mental health issues of the elderly group are becoming increasingly prominent. **Objectives**: The present study aimed to analyze the pathways through which individual, family, and social factors influence Internet use in the elderly through a multi-level analysis framework, to examine the association between Internet use and mental health with a view to providing empirical evidence for digital technology-based mental health intervention programs for the elderly, and to promote the scientific practice of the goal of healthy aging. **Methods**: Based on the data of the 2021 China General Social Survey (CGSS) and provincial Internet development indicators, a mixed cross-sectional dataset was constructed. Logistic hierarchical regression and OLS regression methods were adopted to systematically investigate the multi-level factors associated with Internet use among the elderly group and its association with mental health. **Results**: The results indicate that individual resources (younger age, higher education level, and good health status) and family technical support (family members’ Internet access) are strongly associated with Internet usage among the elderly, while regional Internet penetration rate appears to operate indirectly through micro-mechanisms. Analysis of the association with mental health showed that Internet use was related to a lower score of depressive tendency (*p* < 0.05), and this association remained robust after controlling for variables at the individual, family, and social levels. **Conclusions**: The research results provide empirical evidence for the health promotion policies for the elderly, advocating the construction of a collaborative intervention framework of “individual ability improvement–intergenerational family support–social adaptation for the elderly” to bridge the digital divide and promote the digital integration of the elderly population in China.

## 1. Introduction

At present, China is confronted with the dual structural challenges of an aging population and the digital wave. By the end of 2024, the number of people aged 60 and above in China had exceeded 300 million, accounting for 22.0% of the total population [1]. It is predicted that China will enter a severely aging society by 2035. Meanwhile, the national Internet penetration rate has been continuously rising. The 55th “Statistical Report on the Development of China’s Internet” shows that by the end of 2024, the number of Internet users in China reached 1.108 billion, among which the proportion of those aged 60 and above was only 14.1% [2]. Compared with Internet users of other age groups, the elderly Internet user group lags relatively behind in Internet usage and faces a series of challenges and obstacles [3]. It can be seen that the rapid development of Internet technology is ushering in a new era of information transmission between people [4]. While bringing digital opportunities and dividends to China, it has also brought about a new social governance challenge, namely, the digital divide among the elderly [5]. It may exacerbate the social isolation and health risks of the elderly [6]. The World Health Organization (WHO) points out that insufficient social participation is an important cause of the deterioration of mental health among the elderly, and the Internet, as an important medium for information acquisition and social interaction, has potential associations with mental health in terms of its usage behavior and resulting outcomes [7]. Notably, depression is the most representative mental health issue among the elderly population [8]. Studies have shown that over one-third of the elderly population experience depressive symptoms, and the suicide risk associated with depression is 4 to 5 times higher than that of other age groups. Therefore, this study selects depressive symptoms as the core observation indicator to further explore the potential association between the Internet, as a social connection tool, and mental health.

How to help the elderly integrate into the Internet society and benefit from Internet applications is an important issue. The National Plan for the Development of Aging Affairs during the 14th Five-Year Plan Period clearly proposes the “Smart Assistance for the Elderly” initiative, requiring the narrowing of the digital divide through technological adaptation for the elderly [9]. In April 2021, the Ministry of Industry and Information Technology issued a document to accelerate the special action for the adaptation of Internet applications to the needs of the elderly and the disabled, helping key beneficiary groups such as the elderly to access and use Internet application information equally and conveniently and achieve inclusive social development [10]. Therefore, deepening research on the influencing factors of Internet use among the elderly at multiple levels and exploring the association between Internet use and the mental health of the elderly group is not only a response to national policies but also conducive to identifying obstacles to the elderly’s digital integration and the results of the third-layer digital output gap (i.e., different mental health outcome states), so as to better take measures to actively respond to population aging.

Based on this, this study integrates the 2021 China General Social Survey (CGSS) and 2021 provincial Internet penetration rate data to construct a mixed cross-sectional dataset, aiming to reveal the multi-level influencing factors of Internet use among the elderly group and its association with mental health, providing a scientific basis for helping the elderly bridge the digital divide and integrate into the information society.

## 2. Literature Review and Research Hypotheses

### 2.1. Influencing Factors and Research Hypotheses of Internet Usage Among the Elderly Population

Extensive research exists on the factors influencing Internet use among older adults, both in China and internationally. Overall, it is mainly divided into three categories, including the individual level [11], the family level, and the social level.

At the individual level, age is a significant factor influencing internet use. Younger older adults (typically in their 60s and early 70s) are both more likely to use the internet and use it more frequently than their older counterparts (typically in their late 70s, 80s, and beyond). Compared with men, elderly women are more likely to use the Internet [12]. At the same time, Freese et al. hold that there is a certain correlation between higher educational attainment and income level and the online behavior of the elderly group [13]. In addition, physical health is also one of the key factors affecting the use of the Internet among the elderly, and the Internet access rate of the elderly with visual and hearing impairments is likely to be lower [14]. In view of this, this study further incorporates the physical health level of the elderly into the individual analysis level. Based on this, Hypothesis H1 is proposed: among older adults, those who are younger, female, have higher education levels, have higher personal incomes, and are in better health are more likely to use the internet.

At the family level, the existing research mainly involves three aspects: family structure, social network, and resource conditions. Studies show that marital status, particularly being married, influences internet use among older adults. This is often attributed to the supportive family structure, where spouses learn and help each other navigate online [15]. Meanwhile, elderly people with fewer children and in an empty-nest state are more inclined to use the Internet, which may be related to the concentration of intergenerational technical support [16]. Everett Rogers’ theory of innovation diffusion further points out that when there are Internet users among the relatives or close friends of the elderly, the probability of technology adoption significantly increases, which reflects the demonstration effect of social networks [17]. In terms of resource conditions, the absence of hardware equipment (such as the lack of smartphones or computers) and insufficient operational skills constitute the main obstacles, while the family economic level directly affects the purchase of equipment and the continuous use of network services [18]. These research findings jointly reveal the shaping effects of family size, marital status, social circle technology penetration rate, and resource allocation ability on the digital participation of the elderly group. Based on this, Hypothesis H2 is proposed: The elderly group that is married with a spouse, has fewer children, has other family members surfing the Internet, and has a higher total household income is more likely to use the Internet.

At the social level, the Internet usage behavior of the elderly is significantly influenced by the social environment they are in [19], thereby causing a digital divide between regions [20]. The Internet penetration rate in various regions is positively correlated with the Internet usage rate of the elderly, that is, the elderly in areas with a higher Internet penetration rate are more likely to adopt the Internet [21]. From the perspective of regional distribution, the majority of Internet user groups are concentrated in the eastern region [22]. Against this background, Hypothesis H3 is proposed: The elderly group with a higher Internet penetration rate in the region is more likely to use the Internet.

To sum up, although existing studies have systematically identified characteristic factors at the individual, family, and other levels that influence internet use among older adults, there are two notable limitations. First, most studies adopt a single-level analytical framework, which fails to capture the interaction effects between variables at the individual, family, and social levels, resulting in an incomplete understanding of digital integration mechanisms. Second, insufficient exploration has been conducted on the association between regional digitalization characteristics (e.g., provincial internet coverage) and older adults’ internet usage behaviors in the Chinese context. To address these issues, this study integrates multi-level variables through a hierarchical model, aiming to fill the aforementioned methodological gaps and contextual research deficits.

### 2.2. The Impact of Internet Usage on the Mental Health of the Elderly Group and Research Hypotheses

The double-edged sword effect of Internet usage on the mental health of the elderly group has attracted much attention. On the one hand, the viewpoint of the “Network Gain Effect Theory” holds that the Internet is a convenient and effective means to maintain existing social relationships or establish new ones [23], that is, it has a positive impact on the mental health of the elderly group [24]. Lyu took the elderly aged between 60 and 95 as the research subjects and found that Internet usage was positively correlated with their self-assessed health [25]. By increasing the frequency of contact between the elderly and family members, as well as neighborhood society, more social support was obtained, thereby reducing the sense of alienation. It has a positive impact on improving the mental health of the elderly [26]. On the other hand, studies have also shown that excessive reliance on or improper use of the Internet may also induce anxiety and loneliness in the elderly [27], generate the “presence substitution effect” [28], and further aggravate the depressive tendency of the elderly [29]. Therefore, based on the above content, this study proposes Hypothesis H4: Compared with the elderly group that does not use the Internet, the elderly group that uses the Internet is psychologically healthier.

## 3. Data, Variables, and Methods

### 3.1. Data Sources

The data used in this study consists of two parts. First, this study utilizes data from the Chinese General Social Survey (CGSS). Launched in 2003, the CGSS is China’s first national, multidisciplinary, and longitudinal social survey program. Multi-stage stratified PPS random sampling was adopted to comprehensively collect data at multiple levels, such as individuals, families, and society across the country. A total of 100 county-level units and 5 major cities were selected, which has good representativeness. CGSS2021 is the latest publicly available data, covering 320 communities in 19 provinces, with 8148 valid samples (among the sample population, urban respondents accounted for 56.14%, the proportion of male respondents was 45.15%, the average age was 51.64 years, the average years of education was 9.20 years, and the married population accounted for 73.07%). According to the defined standards of the World Health Organization, this study selected the elderly aged 60 and above as the subjects of analysis and research. After data cleaning, 2929 valid elderly samples were retained. The second is the “2021 China Internet Development Report”, which extracts the Internet application index from it and combines it with CGSS (2021) to form mixed cross-sectional data.

### 3.2. Variable Processing

In the section analyzing the influencing factors of Internet usage among the elderly, the explained variable is whether they use the Internet. The explained variable is Internet usage behavior (binary variable): Based on the response to “How often you used the Internet in the past year”, categorize “rarely”, “sometimes”, “often”, and “always” as “using the Internet” (assigned a value of 1), and categorize “never” as “not using the Internet” (assigned a value of 0). Explanatory variables are divided into three levels, namely, (1) the individual level (including age, gender, years of education, logarithm of personal income, and physical health status), (2) the family level (including marital status, number of children, family members’ Internet usage, and logarithm of total family income), and (3) the social level (referring to the provincial Internet penetration rate).

When analyzing the impact of Internet usage on the mental health of the elderly, this study takes “mental health” as the explained variable. The selection of depressive tendencies as the core measurement indicator is mainly based on the following considerations: First, depressive symptoms are common and representative mental health issues among the elderly, which can effectively reflect the core status of the mental health of the elderly. Second, due to the availability of data from the 2021 China General Social Survey (CGSS), the questionnaire only included measurement questions about feelings of depression (“How often have you felt depressed or frustrated in the past four weeks?”). That is, the A17 question. Although it is a single item, it has been verified for validity by Wang et al. [30] and has been proven to be a proxy indicator for self-assessment of mental health. The measurement results cover an individual’s subjective overall evaluation of their own psychological state. They not only reflect depressive tendencies but also, to a certain extent, common psychological problems such as anxiety and loneliness, comprehensively demonstrating the individual’s mental health status in the recent period. In addition, the selection of this indicator is also based on the availability of data and measurement consistency to ensure the feasibility and reliability of the research analysis. It should be noted that this study acknowledges that a single indicator cannot fully cover all dimensions of mental health. However, under the current data conditions, this indicator can provide effective empirical support for the research hypothesis, and its validity has been verified by existing studies. This question has five options, namely always, often, sometimes, rarely, and never. The five options are assigned values in reverse (that is, always = 4, often = 3, sometimes = 2, rarely = 1, and never = 0). The score range is from 0 to 4. The larger the value, the higher the depression level of the elderly, indicating that the mental health status of the respondents is less optimistic. The explanatory variable is whether the elderly group uses the Internet. The control variables are those at the three levels of individuals, families, and society. The selection and setting of relevant variables are detailed in Table 1.

### 3.3. Research Methods

When exploring the influencing factors of Internet usage among the elderly group in this study, the variable characteristics at the individual, family, and social levels were comprehensively considered. These variables presented significant hierarchical nested relationships in the data structure (such as individuals nested within the province to which they belong). If traditional regression analysis methods (such as ordinary logistic regression) are adopted, their basic assumption requires that the observed samples be independent of each other, but the hierarchical correlation in the actual data will cause the model to violate this assumption. Specifically, ignoring the nested structure between individuals and provinces may cause the following problems: First, the standard errors of the regression coefficients of individual-level variables are underestimated, thereby overestimating the significance level of the test statistics and increasing the risk of the first type of error. Second, failure to distinguish the mechanism of variable effects at different levels may induce “Ecological Fallacy”, that is, mistakenly deducing the statistical relationships at the group level directly to the individual level. Therefore, when traditional regression models analyze data with multi-level characteristics, they not only have difficulty accurately capturing the true correlations among variables but may also lead to misleading conclusions for policy design. Therefore, this study fitted a set of logistic hierarchical regression models and calculated the ICC (Intraclass Correlation Coefficient), which is calculated as ICC = r^2^/(r^2^ + σ^2^), where r^2^ represents the inter-provincial level intercept variance and σ^2^ represents the individual-level residual variance; a threshold of ICC > 5% indicates the necessity of using a multi-level model, by setting a zero model (i.e., without any explanatory variables), and conducted a comparison of information criteria (AIC/BIC) to determine whether to introduce provincial random effects in order to more accurately capture the impact of regional differences on the internet usage of the elderly.

The results show that the estimated value of the random intercept variance at the provincial level is 0.364, and the intra-group correlation coefficient (ICC) is 0.0997 with $\sigma^{2}$ fixed at π^3^/3 ≈ 3.29 in the logistic model, which is significantly higher than the critical value of 5%. This indicates that there is significant inter-provincial heterogeneity in the Internet usage behavior of the elderly group, and 9.97% of the individual differences can be attributed to the structural characteristics at the provincial level. Furthermore, the AIC/BIC value of the multi-layer model is smaller, indicating that its fitting is better. To this end, this study adopted the logistic hierarchical regression model. By including the provincial random intercept term, the problem of sample autocorrelation was effectively corrected, and at the same time, the potential influence effect of regional characteristics on Internet usage behavior could be captured more accurately.

In the model construction stage of the second half of this study, although ordered outcome variables are usually analyzed using nonlinear models (such as ordered Logit), when the outcome variables have five or more ordered classifications and the distribution skewness is not severe, they can be regarded as continuous numerical independent variables for analysis. In this study, the depression score (0–4 points) was classified into 5 categories, and the skewness value was −0.38 (slightly left-skewed), which met the applicable conditions of OLS. Therefore, it is acceptable for this study to quantitatively examine the correlation between Internet usage and the mental health of the elderly by using the ordinary least squares (OLS) regression method. At the same time, to avoid the underestimation or overestimation effect caused by reducing the independent variable to a binary categorical variable and thus losing all levels of Internet usage intensity. In this study, the five categories of frequency of Internet use were retained in the robustness test and re-validated using ordered logit regression.

## 4. Sample Description and Empirical Findings

### 4.1. Sample Analysis

According to the statistical results (Table 2), the average age of the elderly population in the sample is 70.03 years old (standard deviation = 6.81). The gender distribution is relatively balanced, with 50.8% being male and 49.2% being female. The average educational attainment was 2.33 (standard deviation = 1.44), indicating that the overall educational level was concentrated at junior high school and below. In terms of economic conditions, the mean logarithm of individual annual income and the logarithm of household annual income were 8.11 and 9.62, respectively, indicating significant individual differences in income levels. In terms of health status, 70.1% of the elderly self-assessed that they were in good health, while 29.9% had health problems. In terms of marital status, 73.3% of the elderly are married. The average number of children is 2.25, among which the extreme value is the highest, reaching 13 (including adopted children), reflecting the characteristics of some multi-child families. In terms of the Internet usage environment, 65.2% of elderly households have other members using the Internet, and the average Internet penetration rate in the region where they are located is 72.9%. The above results indicate that there is a potential correlation between the Internet usage behavior of the elderly group and their demographic characteristics, family support, and regional digitalization level.

In the process of digitalization, Internet access and usage behaviors are gradually evolving into a new social differentiation mechanism, which gives rise to multi-dimensional segmentation effects on the basis of the existing class structure. Research indicates that based on the differences in the availability of network technology and practical ability, social members can be divided into two major groups: network access users and non-access users [31]. According to the grouping statistics results in Table 3, the Internet user group contains a total of 911 valid samples, and the average mental health level is 0.908 (standard deviation = 1.070). The sample size of the non-Internet user group was 1288, and the mean mental health was 1.273 (standard deviation = 1.192). It can be known through comparison that the mental health indicators of Internet users among the elderly group show a better trend than those of non-users. The significance of the differences between the groups needs to be verified through further statistical tests.

It should be noted that the sample size differences in Table 3 result from the fact that some questionnaire items were not responded to. To maintain the integrity of the data, no interpolation was performed on the missing values.

### 4.2. Empirical Results

#### 4.2.1. Analysis of Influencing Factors for the Use of the Elderly Group

As can be seen from the logistic hierarchical regression results in Table 4, models 2 to 5 systematically reveal the multi-level driving mechanism of Internet usage among the elderly population by gradually incorporating variables at the individual, family, and social levels. Specifically, Model 2, based on the zero model, newly incorporates explanatory variables at the social level. The results show that the coefficient of the provincial Internet penetration rate is 2.944 (*p* < 0.05), indicating a preliminary positive correlation. This provides phased support for H3, but this result only reflects a simple association between the regional digital environment and the Internet usage of the elderly. Furthermore, after adding this variable, the variance of the inter-provincial intercept decreased from 36.4% to 30.5%. This indicates that 5.9% of the differences in Internet usage among the elderly between provinces can be explained by the Internet penetration rate.

Model 3 contains individual-level variables. The odds ratio results (Table 5) show that the age odds ratio is 0.92, meaning that for every additional year of age, the advantage of the elderly in using the Internet drops to 92% of the original. This reflects the natural decline of physiological functions and cognitive abilities with age, thereby affecting technology adoption behavior [32]. The gender advantage ratio is 0.61, indicating that the advantage of elderly women in using the Internet is 1.64 times that of elderly men (1/0.61), supporting the gender difference hypothesis of H1, which may be related to the socialization differences of gender roles [33]. The advantage ratio of educational attainment is 1.74, indicating that for every 1-unit increase in educational level, the advantage of using the Internet increases by 1.74 times, confirming the core viewpoint of the new human capital theory. The years of education reflect an individual’s early accumulation of human capital, which directly affects the efficiency of technology adoption by enhancing cognitive abilities such as information processing and logical reasoning [34]. The odds ratio of the logarithm of personal income is 1.05, indicating that the availability of economic resources can indirectly enhance the advantage of use by lowering the threshold of equipment and service [35]. In addition, the odds ratio of physical health is 1.66, indicating that elderly people with better health are more likely to actively integrate into digital life. The above results fully support Hypothesis H1.

Model 4 further incorporates variables at the family level. Among them, the odds ratio of family members’ Internet access is 4.99, indicating that when there are other Internet users in the family, the advantage of the elderly in using the Internet is 4.99 times that of families without family members accessing the Internet, supporting the core assumption of “family technical support” in H2. However, marital status (odds ratio 1.23), the number of children (odds ratio 0.93), and the logarithm of total household income (odds ratio 1.01) were not significant. This might be because married elderly people are more dependent on offline social interaction [36], and economic conditions are not the key driving factor at the family level. In conclusion, only the “family members’ Internet access situation” in H2 has been verified, while the rest of the hypotheses need further exploration. Furthermore, the inter-provincial variance further decreased to 0.180 (ICC ≈ 5.2%), indicating that the household variable captured additional regional heterogeneity.

After integrating all the variables in Model 5, the odds ratio and significance of the variables at the individual and household levels did not change significantly. However, the odds ratio of the provincial Internet penetration rate dropped to 3.95 (not significant), indicating that its direct effect disappeared. This change is not contradictory but reveals a deeper mechanism of action: the impact of regional Internet penetration rate does not directly affect individuals but is indirectly achieved through micro-level pathways, such as empowering personal resources and strengthening family technical support, which is consistent with the “social network mediating effect” in the theory of innovation diffusion. In conclusion, the “direct correlation between regional Internet penetration rate and Internet usage by the elderly” hypothesized by H3 was not ultimately supported, but its indirect impact generated through micro-mechanisms was verified.

From the perspective of model selection, Model 4 is regarded as the optimal explanatory framework due to the significant core variables, the reasonable reduction in inter-provincial variance, and the better simplicity of the model. The results show that the Internet usage of the elderly group is mainly driven by individual capabilities (low age, high education level, and health) and family support (family members’ Internet access). Regional policies need to be based on micro-mechanisms, giving priority to strengthening educational investment and family digital mutual assistance, rather than relying solely on macro-level digital infrastructure.

#### 4.2.2. Analysis of the Impact of Internet Use on the Mental Health of the Elderly Group

Table 6 presents the regression analysis results of the relationship between Internet usage and mental health among the elderly group. Model (1) only included the core independent variables. The results showed that the regression coefficient of Internet usage was −0.365 (*p* < 0.01), indicating that the preliminary association was statistically significant. After introducing individual-level control variables in Model (2), the absolute value of the Internet usage coefficient decreased to −0.128 (*p* < 0.05), while the explanatory power of the model significantly improved (Adj R^2^ increased from 0.024 to 0.187), suggesting that individual characteristics partially explain the influence mechanism of Internet usage.

In Model (3), which further incorporates family-level variables, the correlation coefficient between internet use and depressive tendency scores was adjusted to −0.108 (*p* < 0.05), with marital status also showing a significant negative association (β = −0.129, *p* < 0.05). Notably, variables such as household income and number of children did not reach statistical significance, indicating that the direct association between family structure factors and mental health may be limited. As shown in Model (4), after controlling for variables at the individual, family, and social levels, internet use among older adults remained significantly negatively associated with depressive tendency scores (β = −0.099, *p* < 0.05), meaning that the average depressive tendency score of older adults who use the internet was 0.099 units lower than that of non-users. In terms of practical association strength, the depressive level of internet users was approximately 7.8% lower than the mean score of non-users (1.273), which is consistent with the “network gain effect”—suggesting an association between internet use and improved mental health through enhanced social connectivity.

In conclusion, although the gradual introduction of control variables led to the attenuation of the Internet usage coefficient, its significance remained stable (*p* < 0.05 or above), confirming a robust association between Internet usage in the elderly population and the improvement in mental health. Combined with the design logic of the scale, the negative coefficient indicates that Internet usage works by reducing the mental health score. Hypothesis H4 is thus supported, that is, using the Internet is beneficial to improving the mental health of the elderly group.

### 4.3. Robustness Test

Since the above analysis may have problems such as endogeneity, variable definition, and model selection, which lead to insufficient robustness of the results, this paper adopts the method of adjusting the control variables to conduct the robustness test. The specific approach is as follows: By using the stepwise regression method, the control variables with the lowest coefficient significance levels are eliminated one by one. Eventually, only the control variables that are significantly non-zero at the 10% level are retained. The regression results are shown in column (1) of Table 7. It can be seen that the significance level of Internet usage has further increased, and the sign of the regression coefficient remains consistent with the benchmark regression result, verifying the robustness of the previous analysis results.

Furthermore, to avoid the loss of usage intensity information caused by simplifying the independent variable (Internet usage) to a binary categorical variable (yes/no), the following analysis was supplemented in the robustness test in this study: First, the original five-level classification of Internet usage frequency was retained as an ordered variable (named Internet use_1). The second is to adopt ordered logit regression instead of OLS to handle the ordered classification dependent variable of mental health (depression score: 0–4 points) more rigorously.

The regression results are shown in Table 8. The regression coefficient of Internet usage frequency is −0.057 (*p* < 0.1), indicating that higher intensity of Internet usage is still significantly associated with a lower tendency towards depression (the coefficient direction is consistent with the benchmark model). Although the significance level slightly declined (affected by the increase in classification granularity), the core conclusion remained robust.

## 5. Conclusions

Based on the 2021 China General Social Survey (CGSS) and provincial internet development data, this study systematically examined the multi-level driving mechanisms of internet use among older adults in China and its association with mental health. The findings are as follows: First, the association between individual resources and internet use. This study found that younger age, higher educational attainment, and better physical health were associated with a higher probability of internet use among older adults. Second, the association between family technical support and internet use. The presence of other internet users in the family was associated with more frequent internet use among older adults, indicating that the technical environment within the family may be an important related factor. However, marital status and number of children did not reach statistical significance. For marital status, this may be because married older adults rely more on offline social interactions; additionally, spouses of married older adults may also have limited digital skills (e.g., both being in an advanced age group), resulting in the absence of mutual learning effects, and their reliance on offline family interactions may reduce their instrumental needs for the internet. For the number of children, the non-significant association with internet use may reflect that the “quality” of filial support (e.g., actual provision of technical guidance) is more critical than “quantity.” Against the backdrop of increasing empty-nest elderly in China, even a larger number of children may not translate into effective support if they live remotely or lack willingness to assist; conversely, families with fewer children but co-residing or engaging in frequent interactions may provide more sustained support. These findings differ from some previous studies, potentially due to differences in sample characteristics, variable measurement, or model covariates. They suggest the context-dependence of family factors in older adults’ digital integration, highlighting the need for future research to refine micro-mechanisms of intra-family technology transfer.

Third, the indirect role of regional internet penetration. The effect of provincial internet penetration weakened after controlling for individual and family variables, indicating that it exerts influence indirectly through empowering micro-mechanisms, which aligns with Chong et al.’s conclusion that regional digitalization must be integrated with local resources. The fourth is the association between Internet usage and mental health, but this association needs to be interpreted from a more complex perspective: On the one hand, this result supports the “network gain effect”, that is, Internet usage is associated with a reduced tendency towards depression in the elderly (β = −0.099, *p* < 0.05). On the other hand, there is a possibility of a two-way relationship between the two, that is, elderly people with better mental health conditions may be more inclined to try Internet technology or have a stronger ability to maintain digital engagement behavior. This bidirectionality has also been reflected in previous studies. For example, the 9-year follow-up study of the MIDUS project found a dynamic interaction between computer use and adult health benefits [37]. Due to the limitations of cross-sectional data and research methods, this study is unable to clearly define the causal direction of the association between the two and can only reflect the statistical correlation between the variables. Future research could adopt longitudinal data or instrumental variable methods to further clarify the complex mechanisms between digital usage and mental health.

## 6. Discussion

### 6.1. Academic Contributions and Theoretical Significance

This study expands the existing literature in the following aspects:(1)Integration of multi-level perspectives. Traditional studies mostly focus on a single level (such as individuals or families), while this study reveals the interaction among individuals, families, and social factors through a hierarchical model; verifies the view that regional Internet infrastructure needs to be coordinated with micro-mechanisms; and provides a new analytical framework for the study of the digital divide.(2)Regarding the exploration of mechanisms underlying the association with mental health, unlike the “presence substitution effect” proposed in existing studies, this research found a robust association between internet use and better mental health status among older adults. This finding provides a theoretical reference for mental health intervention practices targeting the elderly.

### 6.2. Practical Insights and Policy Suggestions

Based on the research conclusions, this study proposes the following multi-level and operational policy recommendations, aiming to systematically bridge the digital divide among the elderly and promote mental health through a collaborative framework of “individual ability enhancement–intergenerational support from families–social adaptation for the elderly”.

(1)Individual ability enhancement: Establishing a full life-cycle education system.

Relying on community service centers, carry out “age-friendly” digital skills training, with a focus on those with low educational levels and the elderly. The teaching content should simplify the operation steps (such as video calls and online medical appointments). Incorporate Internet usage into elderly health promotion projects to form an integrated health management system. For instance, by using smart bracelets to monitor health data and interact with family doctors, the practical perception of technology usage among the elderly can be enhanced.

(2)Intergenerational support within families: Promote the digital mutual assistance model.

Strengthen intergenerational support within families, promote the demonstration project of “Digital Mutual Aid Families”, encourage children to help the elderly master basic operational skills through regular training, and achieve intergenerational technology sharing. For families with a small number of children or those living in scattered areas, a volunteer pairing assistance mechanism should be established, with community workers or college students providing on-site technical guidance.

(3)Social adaptation for the elderly: Optimizing policy synergy.

Optimize regional policy design. While enhancing Internet coverage (such as the construction of 5G base stations in rural areas), provide supporting “home device subsidies” and “data traffic reduction plans” to lower usage costs. Achieve a combination of regional infrastructure construction and micro-empowerment, and avoid the investment deviation of “emphasizing hardware over capability”. Emphasize the design of standardized products suitable for the elderly. It is suggested that mainstream apps (such as Alipay and Douyin) add an “elderly mode”, simplify the interface, and block redundant information to avoid anxiety caused by complex operations.

It should be noted that the above suggestions in this study need to be interpreted in the specific context of China: First, national-level policies such as “Smart Assistance for the Elderly” provide top-level design support for the digital integration of the elderly, which is different from the decentralized policy systems in some countries; second, the traditional multi-generational family structure in China has created natural conditions for the intergenerational transmission of technology, while in a society dominated by nuclear families, this support mechanism may be relatively weak. Thirdly, the digital infrastructure gap caused by China’s unique urban-rural dual structure may lead to more prominent access obstacles for the elderly in rural areas of our country. Therefore, in societies where individualistic culture prevails or the family support function weakens, it is necessary to reduce the weight of intervention at the family level and instead strengthen the community digital support network and universal technology subsidy policies to adapt to different social contexts.

### 6.3. Research Limitations and Future Directions

Although this study reveals the influencing factors of Internet use among the elderly and its mental health effects through multi-level analysis, there are still the following limitations, which can be further improved in multiple dimensions in the future:

First, limitations in data timeliness and causal inference. Based on the 2021 cross-sectional data, this study can identify correlations between variables but fails to capture the dynamic evolution of Internet usage behavior and mental health status. It cannot completely rule out the interference of the “health selection effect” (i.e., elderly individuals with better mental health are more likely to use the Internet actively) or unobserved confounding factors (such as social participation preferences). Future research can adopt a longitudinal tracking design (e.g., relying on the China Health and Retirement Longitudinal Study, CHARLS), use panel data to analyze the life-cycle path of technology adoption, and combine propensity score matching (PSM) or instrumental variable methods to verify causal relationships more rigorously.

Second, insufficient refinement of variable measurement. The current operationalization of “Internet use” is only based on frequency, without distinguishing behavior types (e.g., passive information browsing, active social interaction, and online medical service use) and depth of use (e.g., skill proficiency and functional diversity). Mental health measurement mainly relies on a single item of depressive tendency, failing to comprehensively cover multi-dimensional indicators such as anxiety and loneliness. Subsequent studies can adopt multi-dimensional scales (e.g., the EU IST standard digital literacy scale) to refine the measurement dimensions of Internet use and combine standardized tools such as the Center for Epidemiologic Studies Depression Scale (CES-D) to improve the completeness of mental health assessment, thereby revealing the heterogeneous impact of different digital activities on mental health.

Third, the depth of mechanism analysis needs to be expanded. Although this study verifies the importance of family technical support, it does not explore its specific paths (e.g., emotional motivation vs. instrumental guidance); the indirect effect of regional digital infrastructure also needs to be further explained with more micro-level social interaction data (e.g., neighborhood digital support networks). Future research can integrate quantitative and qualitative methods, through in-depth interviews or participatory observation, to analyze the psychological mechanism of intergenerational digital mutual assistance and the construction logic of community digital inclusive environments.

Finally, the generalizability boundary of research conclusions needs clarification. This study focuses on the Chinese context, and its findings are closely related to local characteristics such as the “Smart Assistance for the Elderly” policy and intergenerational co-residence culture. Future cross-cultural comparative studies (e.g., comparing East Asian and European American countries) can explore differences in elderly digital integration models under different social policies and cultural backgrounds, providing a more adaptable theoretical framework for digital inclusion practices in global aging societies.

## Figures and Tables

**Table 1 healthcare-13-01931-t001:** Explanation of variable meanings.

Variable Name	Assignment Description
Internet use	Yes = 1; No = 0
Mental health	Always = 4, often = 3, sometimes = 2, rarely = 1, and never = 0
Age	Actual age
Gender	0 = female, 1 = male
Educational attainment	Having received no education = 0, private school or literacy class = 1, primary school = 2, junior high school = 3, vocational/general high schools, technical secondary schools, and technical schools = 4, and college, junior college, and above = 5
Logarithm of personal income	Take the logarithm of the individual’s total annual income in 2020
Physical health condition	Very unhealthy, relatively unhealthy = 0, average, relatively healthy, very healthy = 1
Marriage	Others = 0, married with a spouse = 1
The number of children	The total number of sons/daughters (including adopted sons/adopted daughters)
The situation of family members’ Internet access	Not going online = 0, going online = 1
Logarithm of total household income	Take the logarithm of the total household income for the whole year of 2020
Internet penetration rate	Internet penetration data by province

**Table 2 healthcare-13-01931-t002:** Descriptive statistics on the current status of internet use among the elderly.

Variable Name	Mean	SD	Min	Max
Age	70.028	6.813	60	99
Gender	0.508	0.500	0	1
Educational attainment	2.330	1.442	0	5
Logarithm of personal income	8.112	3.639	0	16.117
Physical health condition	0.701	0.458	0	1
Marriage	0.733	0.442	0	1
The number of children	2.254	1.270	0	13
The situation of family members’ Internet access	0.652	0.477	0	1
Logarithm of total household income	9.618	2.945	0	16.118
Internet penetration rate	0.729	0.081	0.574	0.919

**Table 3 healthcare-13-01931-t003:** Descriptive statistics of Internet use groups for the elderly.

Variable Name	Not Using the Internet			Use of the Internet		
	N	Mean	SD	N	Mean	SD
Mental health	1288	1.273	1.192	911	0.908	1.070
Age	1288	71.589	6.912	911	67.774	5.965
Gender	1288	0.512	0.500	911	0.503	0.500
Educational attainment	1280	1.849	1.352	906	3.011	1.288
Logarithm of personal income	1288	7.459	3.751	944	9.050	3.250
Physical health condition	1286	0.631	0.483	911	0.800	0.400
Marriage	1288	0.699	0.459	911	0.783	0.413
The number of children	1288	2.516	1.341	911	1.877	1.052
The situation of family members’ Internet access	1225	0.497	0.500	887	0.868	0.339
Logarithm of total household income	1288	9.095	3.103	911	10.381	2.492
Internet penetration rate	1288	0.719	0.079	911	0.744	0.082

**Table 4 healthcare-13-01931-t004:** Logistic stratified regression model of influencing factors of Internet use among the elderly in China.

Variables	Model 1	Model 2	Model 3	Model 4	Model 5
Age			−0.085 ***	−0.068 ***	−0.068 ***
			(0.008)	(0.010)	(0.010)
Gender			−0.488 ***	−0.433 ***	−0.427 ***
			(0.108)	(0.117)	(0.117)
Educational attainment			0.554 ***	0.503 ***	0.502 ***
			(0.045)	(0.047)	(0.047)
Logarithm of personal income			0.051 ***	0.038 *	0.037 *
			(0.016)	(0.020)	(0.020)
Physical health condition			0.508 ***	0.447 ***	0.443 ***
			(0.117)	(0.125)	(0.125)
Marriage				0.211	0.206
				(0.132)	(0.132)
The number of children				−0.069	−0.064
				(0.054)	(0.054)
The situation of family members’ Internet access				1.608 ***	1.608 ***
				(0.129)	(0.129)
Logarithm of total household income				0.008	0.008
				(0.026)	(0.026)
Internet penetration rate		2.944 **			1.374
		(1.600)			(1.326)
The intercept term	−0.441 ***	−2.541 **	3.666 ***	1.535 **	0.574
	(0.143)	(1.148)	(0.607)	(0.707)	(1.128)
Inter-provincial intercept variance	0.364 ***	0.305 ***	0.215 ***	0.180 ***	0.164 ***
	(0.129)	(0.028)	(0.087)	(0.080)	(0.076)

Standard errors in parentheses (*** *p* < 0.01, ** *p* < 0.05, * *p* < 0.1).

**Table 5 healthcare-13-01931-t005:** Logistic hierarchical regression result: odds ratio.

Variable Name	(1)	(2)	(3)
	Odds Ratio	Odds Ratio	Odds Ratio
Age	0.92 ***	0.93 ***	0.93 ***
	[0.90,0.93]	[0.92,0.95]	[0.92,0.95]
Gender	0.61 ***	0.65 ***	0.65 ***
	[0.50,0.76]	[0.52,0.82]	[0.52,0.82]
Educational attainment	1.74 ***	1.65 ***	1.65 ***
	[1.59,1.90]	[1.51,1.81]	[1.51,1.81]
Logarithm of personal income	1.05 **	1.04	1.04
	[1.02,1.09]	[1.00,1.08]	[1.00,1.08]
Physical health condition	1.66 ***	1.56 ***	1.56 ***
	[1.32,2.09]	[1.22,2.00]	[1.22,1.99]
Marriage		1.23	1.23
		[0.95,1.60]	[0.95,1.59]
The number of children		0.93	0.94
		[0.84,1.04]	[0.84,1.04]
The situation of family members’ Internet access		4.99 ***	4.99 ***
		[3.88,6.43]	[3.88,6.43]
Logarithm of total household income		1.01	1.01
		[0.96,1.06]	[0.96,1.06]
Internet penetration rate			3.95
			[0.29,53.12]
var(_cons)	1.24 *	1.20 *	1.18 *
	[1.05,1.47]	[1.02,1.40]	[1.02,1.37]

Exponentiated coefficients; 95% confidence intervals in brackets (*** *p* < 0.01, ** *p* < 0.05, and * *p* < 0.1).

**Table 6 healthcare-13-01931-t006:** Results of Internet use and mental health return in the elderly population in China.

Variables	(1)	(2)	(3)	(4)
	Mental Health	Mental Health	Mental Health	Mental Health
Internet use	−0.365 ***	−0.128 **	−0.108 **	−0.099 *
	(0.049)	(0.051)	(0.054)	(0.054)
Age		−0.005	−0.009 **	−0.008 **
		(0.003)	(0.004)	(0.003)
Gender		−0.236 ***	−0.248 ***	−0.267 ***
		(0.046)	(0.047)	(0.048)
Educational attainment		−0.072 ***	−0.072 ***	−0.067 ***
		(0.018)	(0.019)	(0.019)
Logarithm of personal income		−0.019 ***	−0.007	−0.005
		(0.007)	(0.008)	(0.008)
Physical health condition		−0.874 ***	−0.847 ***	−0.837 ***
		(0.050)	(0.051)	(0.051)
Marriage			−0.129 **	−0.121 **
			(0.054)	(0.054)
The number of children			0.030	0.020
			(0.020)	(0.020)
The situation of family members’ Internet access			−0.040	−0.041
			(0.052)	(0.052)
Logarithm of total household income			−0.016	−0.016
			(0.010)	(0.010)
Internet penetration rate				−0.839 ***
				(0.287)
The intercept term	1.273 ***	2.603 ***	2.971 ***	3.507 ***
	(0.032)	(0.255)	0.280	(0.334)
Adj R^2^	0.024	0.187	0.195	0.198

Standard errors in parentheses (*** *p* < 0.01, ** *p* < 0.05, * *p* < 0.1).

**Table 7 healthcare-13-01931-t007:** Robustness test results: the stepwise regression method.

Variable Name	(1)
	Mental Health
Internet use	−0.129 **
	(0.049)
Controlled variable	Controlled
Adj R^2^	0.191

Standard errors in parentheses (** *p* < 0.05).

**Table 8 healthcare-13-01931-t008:** Robustness test results: ordered logit regression.

Variables	(1)
	Mental Health
Internet use_1	−0.057 *
	(0.032)
Controlled variable	Controlled
Adj R^2^	0.079

Standard errors in parentheses (* *p* < 0.1).

## Data Availability

The data presented in this study are available on request from the corresponding author (Xinying He).

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
