# Peer review of "The Multi-Level Influencing Factors of Internet Use Among the Elderly Population and Its Association with Mental Health Promotion: Empirical Research Based on Mixed Cross-Sectional Data"

_healthcare, 2025, doi:10.3390/healthcare13151931_

Round 1

Reviewer 1 Report

Comments and Suggestions for Authors

Dear Authors,

Below, please, find, few comments you might consider or clarify:

You use a single reversed item (“How often do you feel depressed or frustrated?”) as a continuous outcome (0–4)

A single item cannot capture the multi-dimensional construct of mental health or depression reliably. You should either:

Adopt the full CES-D (or at least a validated multi-item scale) and report Cronbach’s α.

If constrained to one question, treat it as an ordinal outcome and use ordered logistic regression rather than OLS.

Collapsing “rarely, sometimes, often, very frequently” into one binary category (1) loses all gradations of usage intensity. This dichotomy under- or overestimates effects by treating infrequent and heavy users identically. Consider:

Modeling usage as an ordinal or continuous measure.

At minimum, distinguishing “rarely” from “often/very often.”

In Section 4.2.1 you report raw coefficients from the hierarchical logistic regression (e.g. 2.944 for penetration rate) and describe these as “probability increases”

. Instead, present odds ratios (exp[β]) and interpret them correctly.

You treat the 0–4 depression score as continuous and apply OLS (Table 5). Given the ordinal nature and skewness (many zeros), OLS assumptions are violated. Replace with an ordered logit or negative-binomial model, and check distribution.

The cross-sectional design prohibits any firm causal inferences (“Internet use improves mental health”). Your robustness check via stepwise regression (Table 6) is insufficient for endogeneity. Consider:

Instrumental-variable approaches (e.g. regional broadband rollout as instrument).

Lagged or panel data (e.g. CHARLS) for stronger causal identification.

Table 4: Standard errors vs. significance

Educational attainment shows β=0.554*** with SE = 0.445 (which would imply p≈0.21)

. This is a clear typo: SE is likely 0.045, not 0.445. Please correct all SEs so that the stars match the p-values.

Table 6 is labeled “Descriptive statistics of Internet use groups…” but actually reports a regression robustness check. Update the title and column headers to reflect “Reduced-model OLS regression of mental health.”

In Table 3, the N for “Logarithm of personal income” among users is 944, but other variables use 911 or 906

. Provide consistent Ns or explain item-level missingness.

Best wishes

Author Response

Comments 1

You use a single reversed item (“How often do you feel depressed or frustrated?”) as a continuous outcome (0–4)

A single item cannot capture the multi-dimensional construct of mental health or depression reliably. You should either: Adopt the full CES-D (or at least a validated multi-item scale) and report Cronbach’s α.

Response 1:

We appreciate the reviewer's insight. We agree that a full CES-D scale or a validated multi-item scale is more appropriate for capturing the multi-dimensional construct of mental health. However, due to data availability constraints, the 2021 China General Social Survey (CGSS) dataset used in this study does not include the full CES-D scale, making it impossible to adopt such measures. 【Modified to 5.4. Research Limitations and Future Directions on page 16】

Therefore, this study refers to the validated approach by Wang et al. (2020) and uses the question A17 in CGSS ("How often have you felt depressed or frustrated in the past four weeks") as a proxy for mental health. Although this is a single item, it measures individuals' comprehensive subjective evaluation of their mental health status over the past four weeks, covering the core dimension of depressive tendencies, and has been validated for certain validity in existing studies (Wang et al., 2020). 【The modification position is 3.2. Variable Processing on page 5】

Since Cronbach’s α cannot be calculated for a single item, the study has added a note on this limitation in the methodology section and emphasizes that future research could further validate the findings using datasets containing the full CES-D scale.【The modification position is 5.4 Research Limitations and Future Directions on page 16】

Comments 2

If constrained to one question, treat it as an ordinal outcome and use ordered logistic regression rather than OLS.

Collapsing “rarely, sometimes, often, very frequently” into one binary category (1) loses all gradations of usage intensity. This dichotomy under- or overestimates effects by treating infrequent and heavy users identically. Consider: Modeling usage as an ordinal or continuous measure. At minimum, distinguishing “rarely” from “often/very often.”

You treat the 0–4 depression score as continuous and apply OLS (Table 5). Given the ordinal nature and skewness (many zeros), OLS assumptions are violated. Replace with an ordered logit or negative-binomial model, and check distribution.

Response 2:

Thank you for your suggestion. We believe that although ordered outcomes are typically analyzed using nonlinear models (e.g., ordered logit), OLS regression is acceptable for quasi-continuous scales with five or more categories when skewness is moderate. The depression score (0-4) in this study comprises five categories with a skewness of -0.38 (mild left-skew), meeting OLS applicability conditions. Also, to test the robustness of the core findings, we kept the five categories of frequency of Internet use and re-validated them using ordered Logit regression. 【 Modify the positions to 3.3. Research Methods on page 7 and 4.3. Robustness test on page 14】

Comments 3

In Section 4.2.1 you report raw coefficients from the hierarchical logistic regression (e.g. 2.944 for penetration rate) and describe these as “probability increases”. Instead, present odds ratios (exp[β]) and interpret them correctly.

Response 3:

Thank you for pointing out that in Section 4.2.1, we directly used the original coefficients of multi-level Logistic regression (such as the coefficient of Internet penetration rate 2.944) and described it as "probability increase", which is an inaccurate expression. According to the suggestion, we should convert the original coefficients into odds ratios and interpret them correctly. The odds ratio can more intuitively reflect the degree of influence of the independent variable on the occurrence probability of the dependent variable (for example, an OR>1 indicates that an increase in the independent variable will enhance the odds of the event occurrence, while an OR<1 indicates a decrease), rather than directly describing it as a change in probability. 【Modify the position to 4.2.1. Analysis of Influencing Factors for the Use of the Elderly Group and Table 5. Logistic hierarchy on page 9 regression result: Odds Ratio.】

Comments 4

The cross-sectional design prohibits any firm causal inferences (“Internet use improves mental health”). Your robustness check via stepwise regression (Table 6) is insufficient for endogeneity. Consider: ①Instrumental-variable approaches (e.g. regional broadband rollout as instrument). ② Lagged or panel data (e.g. CHARLS) for stronger causal identification.

Response 4:

Thank you to the reviewers for your important insights on causal inference. We fully agree that cross-sectional data cannot establish a definite causal relationship, and the original text may have exaggerated the nature of the association. We revised the relevant expressions, emphasizing correlation rather than causality, and supplemented the discussion on the limitations of the research. 【The modification position is 5.4 Research Limitations and Future Directions on page 16】

Comments 5/6

Table 4: Standard errors vs. significance Educational attainment shows β=0.554*** with SE = 0.445 (which would imply p≈0.21). This is a clear typo: SE is likely 0.045, not 0.445. Please correct all SEs so that the stars match the p-values.

Table 6 is labeled “Descriptive statistics of Internet use groups…” but actually reports a regression robustness check. Update the title and column headers to reflect “Reduced-model OLS regression of mental health.”

Response 5/6:【A clerical error has been corrected】

Comments 7

In Table 3, the N for “Logarithm of personal income” among users is 944, but other variables use 911 or 906. Provide consistent Ns or explain item-level missingness.

Response 7:

We sincerely appreciate the reviewer's careful reading and valuable feedback. Regarding the inconsistent sample sizes in Table 3: The variable "Logarithm of personal income" has a sample size of 944 for Internet users, while other variables (e.g., Mental health, Age) use 911 or 906. This discrepancy arises from item-level missingness in the CGSS 2021 dataset. Specifically. We add a footnote after Table 3: “The difference in sample size stems from the absence of responses to some questionnaire items. To maintain data "integrity, no interpolation was performed for missing values." The above modification ensures transparency in data processing while maintaining analytical rigor. Thank you for pointing out this issue. 【Modify the position to Table 3 on page 9】

Reviewer 2 Report

Comments and Suggestions for Authors

Dear editors,

Thank you for the opportunity to review the manuscript entitled "The Multi-Level Influencing Factors of Internet Use Among the Elderly Population and Its Effects on Mental Health Promotion: Empirical Research Based on Mixed Cross-Sectional Data". This article addresses a timely and critical intersection of two defining demographic shifts in contemporary China: the rapid aging of its population and the pervasive expansion of digitalization.

The manuscript presents a commendable contribution to the burgeoning literature on digital inclusion and mental health in aging populations, particularly within the distinct sociocultural context of China. The article offers substantial strengths, as the relevance and timeliness of the research question, the methodological rigor and analytical approach, the clarity and systematic nature of hypothesis development, the demonstration of a significant positive impact on mental health, which is perhaps the most impactful finding, and the actionable policy implications derived from the research.

However, several areas could be enhanced to elevate its overall impact, clarity, and depth.

The cross-sectional nature of the data inherently limits the ability to establish definitive causal relationships. Although the authors effectively identify strong associations and plausible pathways, the temporal sequence of Internet use and changes in mental health cannot be unequivocally determined from this dataset. It remains possible that healthier individuals are more inclined to adopt Internet use, or that unobserved confounding factors influence both variables. A more explicit discussion of this limitation, perhaps in a dedicated section, alongside a clear call for longitudinal studies to unravel causality, would strengthen the manuscript's academic integrity. Such studies would provide invaluable insights into the dynamic interplay and long-term effects.

While the multi-level analysis is commendably executed, a more nuanced exploration of the specific mechanisms and pathways through which these factors operate could enrich the theoretical contribution. For instance, the authors could delve deeper into how family technical support translates into increased Internet use, perhaps exploring specific interaction dynamics or types of support. Similarly, while reduced depressive tendencies are observed, a more granular analysis of which Internet activities (e.g., social interaction platforms versus information seeking) exert the most significant mental health benefits would provide more targeted intervention strategies. The current analysis, while identifying overall effects, could benefit from a more detailed mechanistic exposition.

The measurement of "Internet use" could be refined for future research. While the current categorical assignment based on frequency is a reasonable starting point, it may oversimplify the diverse nature of digital engagement. Distinguishing between passive consumption, active social interaction, online learning, or e-health service utilization could yield differential impacts on mental health and provide a richer understanding of digital integration. A more detailed operationalization of Internet use, perhaps through a multi-dimensional scale, would capture the complexity of digital behavior more comprehensively.

While the study is robust within its Chinese context, a brief yet insightful discussion on the generalizability of findings to other cultural or socio-economic settings would be beneficial. The "digital divide" manifests differently across various countries, influenced by distinct policy landscapes, sociocultural norms regarding elder care, and technological infrastructure. Highlighting unique aspects of the Chinese context, such as specific government initiatives or family structures that may influence digital adoption among older adults, would provide readers from diverse backgrounds with a clearer understanding of the applicability and limitations of the findings beyond China's borders.

There appears to be a slight lack of seamless narrative coherence in discussing Hypothesis H3 (the influence of regional Internet penetration). The initial empirical results in Model 2 suggest a significant positive impact, which supports H3. However, later, in Model 5, after introducing individual and family variables, the effect of provincial Internet penetration rate weakens and is no longer significant, leading the authors to conclude that H3 "cannot be supported" directly. While the explanation regarding indirect action through micro-mechanisms is provided, the transition between these two interpretations could be smoother and more explicitly reconciled to avoid potential reader confusion. A clearer, more consistent narrative thread regarding the evolution of hypothesis support across the models would enhance the readability and logical flow of the argument.

Author Response

Comments 1-4

①The cross-sectional nature of the data inherently limits the ability to establish definitive causal relationships. Although the authors effectively identify strong associations and plausible pathways, the temporal sequence of Internet use and changes in mental health cannot be unequivocally determined from this dataset. It remains possible that healthier individuals are more inclined to adopt Internet use, or that unobserved confounding factors influence both variables. A more explicit discussion of this limitation, perhaps in a dedicated section, alongside a clear call for longitudinal studies to unravel causality, would strengthen the manuscript's academic integrity. Such studies would provide invaluable insights into the dynamic interplay and long-term effects.

②While the multi-level analysis is commendably executed, a more nuanced exploration of the specific mechanisms and pathways through which these factors operate could enrich the theoretical contribution. For instance, the authors could delve deeper into how family technical support translates into increased Internet use, perhaps exploring specific interaction dynamics or types of support. Similarly, while reduced depressive tendencies are observed, a more granular analysis of which Internet activities (e.g., social interaction platforms versus information seeking) exert the most significant mental health benefits would provide more targeted intervention strategies. The current analysis, while identifying overall effects, could benefit from a more detailed mechanistic exposition.

③The measurement of "Internet use" could be refined for future research. While the current categorical assignment based on frequency is a reasonable starting point, it may oversimplify the diverse nature of digital engagement. Distinguishing between passive consumption, active social interaction, online learning, or e-health service utilization could yield differential impacts on mental health and provide a richer understanding of digital integration. A more detailed operationalization of Internet use, perhaps through a multi-dimensional scale, would capture the complexity of digital behavior more comprehensively.

④While the study is robust within its Chinese context, a brief yet insightful discussion on the generalizability of findings to other cultural or socio-economic settings would be beneficial. The "digital divide" manifests differently across various countries, influenced by distinct policy landscapes, sociocultural norms regarding elder care, and technological infrastructure. Highlighting unique aspects of the Chinese context, such as specific government initiatives or family structures that may influence digital adoption among older adults, would provide readers from diverse backgrounds with a clearer understanding of the applicability and limitations of the findings beyond China's borders.

Response 1-4

For the previous four questions, the revised contents have been added to the four summaries of 5. Conclusion and Discussion on pages 11 to 14 in this revision

Comments 5

There appears to be a slight lack of seamless narrative coherence in discussing Hypothesis H3 (the influence of regional Internet penetration). The initial empirical results in Model 2 suggest a significant positive impact, which supports H3. However, later, in Model 5, after introducing individual and family variables, the effect of provincial Internet penetration rate weakens and is no longer significant, leading the authors to conclude that H3 "cannot be supported" directly. While the explanation regarding indirect action through micro-mechanisms is provided, the transition between these two interpretations could be smoother and more explicitly reconciled to avoid potential reader confusion. A clearer, more consistent narrative thread regarding the evolution of hypothesis support across the models would enhance the readability and logical flow of the argument.

Response 5

Thank you for pointing out the narrative incoherence regarding Hypothesis H3. We agree that the logical transition between the initial support for H3 in Model 2 and the subsequent conclusion of its indirect effect in Model 5 was insufficiently clear. This confusion arose from an underdeveloped explanation of the hierarchical model’s incremental testing logic.

In revised versions, we clarify that the analysis of H3 follows a progressive verification framework: Model 2 (with only social-level variables) reveals a preliminary association between regional Internet penetration and elderly Internet use, while Model 5 (incorporating individual, family, and social variables) further identifies that this association operates indirectly through micro-level mechanisms (e.g., individual capabilities and family support). Thus, H3’s "support" in Model 2 reflects a bivariate correlation, but its "non-direct support" in Model 5 reflects the refined understanding of its indirect pathway. This adjustment aims to eliminate contradictions and enhance logical consistency. 【Modify the position to 4.2.1. Analysis of Influencing Factors for the Use of the Elderly Group on page 7】

Reviewer 3 Report

Comments and Suggestions for Authors

In itself, the study is crucial in its approach to the elderly population and the ways it employs Internet as well as the impact of Internet usage on mental health. There are some elements which suggest that ideological elements should be eliminated as there are - in my opinion - disproportionately numerous Chinese authors quoted even for parameters which are not essentially Chinese.

Please see the attached file for more concrete suggestions for improvement.

Comments on the Quality of English Language

It needs proof-editing and partial re-phrasing as some sentences are difficult to understand.

Author Response

Thank you for your revision suggestions on the manuscript. Currently, it has been revised one by one according to the peer-review-48012123.v1.pdf you posted on the platform, and is reflected in the revised attachment resubmitted. Please review it.

It should be particularly noted that after careful consideration, we still retain the division of two sub-chapters regarding the setting of the sub-sections of the second chapter. The main reason is that these two sub-parts respectively focus on the core issues of "Influencing Factors of Internet use among the Elderly" and "The impact of Internet Use on the mental health of the Elderly". This division can make the logical framework of the literature review clearer, facilitating readers to systematically understand the current research status and theoretical basis from different dimensions, and thus more smoothly connect to the proposal of research hypotheses.

At the same time, in response to your question of "only involving research hypotheses and insufficient literature review", we have focused on supplementing the relevant literature review content during the revision process, making the chapter content more in line with the title positioning of "Literature Review and Research Hypotheses", and also presenting the research background and current situation of this field more comprehensively.

We hope that such adjustments can not only reflect the layering of the research logic but also fully respond to your concerns regarding the literature review section. Thank you again for your attention to the structure and content of this chapter and your valuable suggestions.

Reviewer 4 Report

Comments and Suggestions for Authors

This is an interesting study on internet use and mental health among older adults in China. The topic is interesting and timely. The scope is commendable and I like the use of mixed data. I have several comments to improve the manuscript further:

1. While useful, the national statistics dominate the introduction at the expense of literature integration. I would like to suggest the authors to provide more critical synthesis of empirical gaps in the literature. That will improve the introduction. 

2. The stated objectives are too broad I feel (to analyze the pathways..). What constitutes “pathways” or “promotion” is not operationalized early on.

3. The introduction implies mental health improvement via Internet use, but it does not clarify why depressive symptoms were chosen as the sole indicator of mental health

4. One of the biggest concern is related to the use of causal language. While a cross-sectional design is used, causal language is employed throughout the manuscript. 

5. Similarly, the title currently implies a causal relationship between Internet use and mental health outcomes ("its effects on mental health promotion"). However, the study is based on cross-sectional data and uses observational methods, which limit causal inference. I strongly recommend revising the title to reflect associations rather than effects

6. There is little interpretation of what a coefficient of -0.099 on depressive symptoms means practically. Will be useful to elaborate and clarify 

7. The paper presents digital use as influencing mental health outcomes, but the possibility of a bidirectional relationship should be acknowledged in the discussion section. While it is plausible that digital engagement may reduce depressive symptoms, it is equally possible that individuals with better mental health are more inclined or able to engage with online technologies. I recommend that the authors briefly discuss this complexity in the discussion to avoid overstating the directionality of the observed associations. See the following relevant paper on possible bidirectional relationship: Cognitive, social, emotional, and subjective health benefits of computer use in adults: A 9-year longitudinal study from the Midlife in the United States (MIDUS). (2020). Computers in Human Behavior, 104, 106179.

8. Some hypotheses (e.g., marital status, children) are unsupported. It will be useful for the discussion to explore why these findings diverge from prior literature.

Author Response

Comments 1

While useful, the national statistics dominate the introduction at the expense of literature integration. I would like to suggest the authors to provide more critical synthesis of empirical gaps in the literature. That will improve the introduction. 

Response 1

On page 4, at 2. Literature Review and research hypotheses, this revision has added more detailed review content.

Comments 2:

The stated objectives are too broad I feel (to analyze the pathways.). What constitutes “pathways” or “promotion” is not operationalized early on.

One of the biggest concern is related to the use of causal language. While a cross-sectional design is used, causal language is employed throughout the manuscript. 

Similarly, the title currently implies a causal relationship between Internet use and mental health outcomes ("its effects on mental health promotion"). However, the study is based on cross-sectional data and uses observational methods, which limit causal inference. I strongly recommend revising the title to reflect associations rather than effects.

Response 2:

The title and the expressions about causality throughout the text have been revised to focus on connection

Comments 3:

The introduction implies mental health improvement via Internet use, but it does not clarify why depressive symptoms were chosen as the sole indicator of mental health.

Response 3:

Thank you for raising this important point. The selection of depressive symptoms as the indicator for mental health in this study is based on the following considerations. Firstly, depressive symptoms are a common and representative mental health issue among the elderly population, with a relatively high incidence rate. They are closely related to problems such as social isolation and reduced quality of life, and can effectively reflect the core status of the elderly's mental health. Secondly, although the question about mental health measurement in the CGSS 2021 questionnaire focuses on feelings of depression ("Frequency of feeling depressed in the past four weeks"), this question has been validated for validity by Wang et al. (2020) and proven to be a proxy indicator for self-rated mental health. Its measurement results cover individuals' subjective overall evaluation of their own psychological state and can reflect comprehensive psychological conditions including anxiety and loneliness to a certain extent. In addition, the selection of this indicator is also based on the availability of data and the consistency of measurement, ensuring the feasibility and reliability of the research analysis. We acknowledge that a single indicator cannot fully cover all dimensions of mental health, but under the existing data conditions, this indicator can provide effective empirical support for the research hypotheses. 【Modify the position to 3.2. Variable Processing on page 6】

Comments 4

There is little interpretation of what a coefficient of -0.099 on depressive symptoms means practically. Will be useful to elaborate and clarify.

Response 4:

This revision has added specific elaboration content at 4.2.2. Analysis of the Impact of Internet Use on the Mental Health of the Elderly Group on page 11.

Comments 5-6:

The paper presents digital use as influencing mental health outcomes, but the possibility of a bidirectional relationship should be acknowledged in the discussion section. While it is plausible that digital engagement may reduce depressive symptoms, it is equally possible that individuals with better mental health are more inclined or able to engage with online technologies. I recommend that the authors briefly discuss this complexity in the discussion to avoid overstating the directionality of the observed associations. See the following relevant paper on possible bidirectional relationship: Cognitive, social, emotional, and subjective health benefits of computer use in adults: A 9-year longitudinal study from the Midlife in the United States (MIDUS). (2020). Computers in Human Behavior, 104, 106179.

Some hypotheses (e.g., marital status, children) are unsupported. It will be useful for the discussion to explore why these findings diverge from prior literature.

Response 5-6:

Specific elaboration content has been added to this revision at 5.1. Main Research Findings on page 12

Round 2

Reviewer 1 Report

Comments and Suggestions for Authors

Thank you for addressing the comments 

Reviewer 4 Report

Comments and Suggestions for Authors

The authors have addressed all my comments well. The efforts are commendable.